# Active energy compression of a laser-plasma electron beam

P. Winkler[1✉], M. Trunk[1], L. Hübner[1,2], A. Martinez de la Ossa[1], S. Jalas[1], M. Kirchen[1], I. Agapov[1], S. A. Antipov[1], R. Brinkmann[1], T. Eichner[1], A. Ferran Pousa[1], T. Hülsenbusch[1], G. Palmer[1], M. Schnepp[2], K. Schubert[1], M. Thévenet[1], P. A. Walker[1], C. Werle[1], W. P. Leemans[1,2] & A. R. Maier[1✉]

Radio-frequency (RF) accelerators providing high-quality relativistic electron beams are an important resource enabling many areas of science, as well as industrial and medical applications. Two decades ago, laser-plasma accelerators[1] that support orders of magnitude higher electric fields than those provided by modern RF cavities produced quasi-monoenergetic electron beams for the first time[2–4]. Since then, high-brightness electron beams at gigaelectronvolt (GeV) beam energy and competitive beam properties have been demonstrated from only centimetre-long plasmas[5–9], a substantial advantage over the hundreds of metres required by RF-cavity-based accelerators. However, despite the considerable progress, the comparably large energy spread and the fluctuation (jitter) in beam energy still effectively prevent laser-plasma accelerators from driving real-world applications. Here we report the generation of a laser-plasma electron beam using active energy compression, resulting in a performance so far only associated with modern RF-based accelerators. Using a magnetic chicane, the electron bunch is first stretched longitudinally to imprint an energy correlation, which is then removed with an active RF cavity. The resulting energy spread and energy jitter are reduced by more than an order of magnitude to below the permille level, meeting the acceptance criteria of a modern synchrotron, thereby opening the path to a compact storage ring injector and other applications.

In a laser-plasma accelerator[1], the interaction of a high-intensity laser pulse with a plasma creates a trailing density modulation, the plasma wave, which supports electric fields several orders of magnitude larger than those provided by modern RF accelerator cavities. Correctly controlled, the plasma wave can trap electrons from the plasma background and then accelerate a well-confined phase-space volume, resulting in a highly relativistic, high-brightness electron beam from only a centimetre-scale plasma.

The field has seen rapid progress in recent years. A series of landmark experiments demonstrated advanced concepts to generate and characterize high-brightness beams[10–15], new laser guiding concepts have extended the interaction length of laser and plasma to result in electron beams of GeV and higher energies[5–9], first steps towards continuous operation have been made[16] and, recently, the long anticipated first gain from a plasma-driven free-electron laser was reported[17].

However, the reproducibility and stability of today's laser-plasma accelerators are still less developed than those of modern RF machines. This can be linked to the micrometre-scale size of the plasma cavity, which leads to extreme accelerating fields and inherently short femtosecond electron bunch durations, but also makes it very challenging to precisely control the injection and acceleration process. Also, a new plasma cavity is created with every laser shot. Because the plasma cavity is essentially formed by the radiation pressure of the laser, even subtle

variations of the drive pulse can result in a modified plasma cavity, thus changing the acceleration fields and dynamics.

The resulting percent-level energy spread and energy jitter typically associated with laser-plasma electron beams are particularly damaging and still effectively prevent laser-plasma accelerators from becoming a viable alternative accelerator technology. For example, free-electron lasers require permille-level energy spread beams[18], whereas injectors for synchrotron light sources have a tight one-percent-level energy acceptance[19,20]. More generally, transporting large energy spread beams from the accelerator to an application can result in adverse chromatic effects that quickly degrade the beam quality available at the interaction point.

To address these challenges, tailored drive laser systems[21] and the deployment of active stabilization techniques[22–24] are expected to improve the performance of future laser-plasma accelerators, but the implementation of these concepts remains challenging.

A more fundamental approach is to improve the spectral properties of laser-plasma electron beams by exploiting their inherent short bunch duration and high peak currents.

For example, a decompression technique was proposed[25,26], which stretches the bunch longitudinally to introduce an energy–position correlation (chirp) and thereby locally reduces the energy spread at the expense of peak current. This technique has enabled the demonstration

[1]Deutsches Elektronen-Synchrotron DESY, Hamburg, Germany. [2]Department of Physics, University of Hamburg, Hamburg, Germany. ✉e-mail: paul.winkler@desy.de; andreas.maier@desy.de

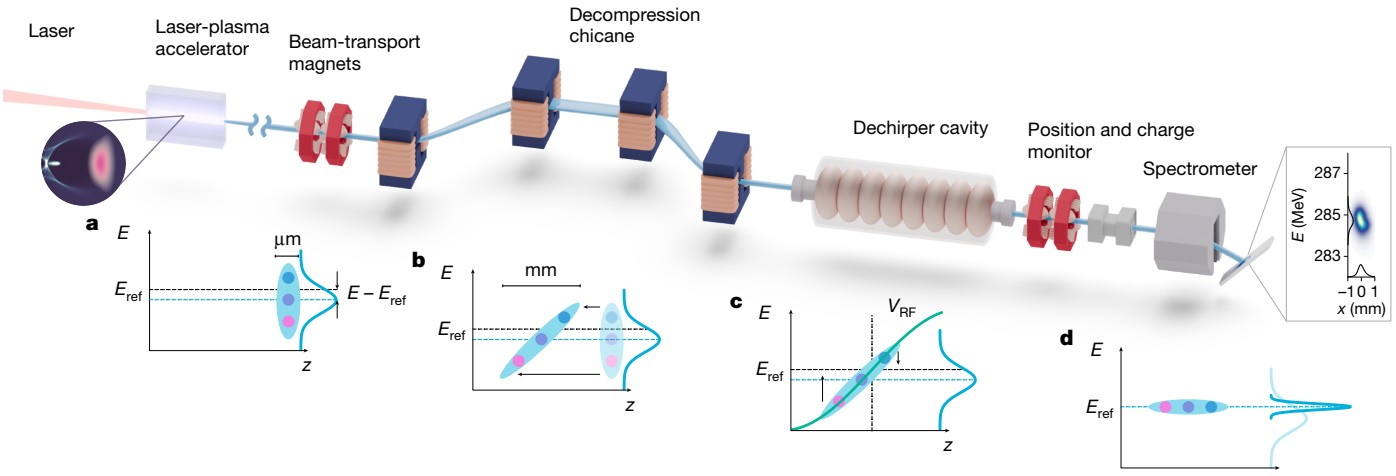

**Fig. 1 | Active energy compression concept. a**, The laser-plasma accelerator provides a several-femtosecond duration electron bunch that has a several-percent energy spread and an energy deviation from the reference energy $E_{ref}$. **b**, In a subsequent magnetic chicane, energy-dependent path-length differences result in a longitudinal energy chirp, which effectively stretches the bunch from micrometre to millimetre length. **c,d**, The RF field of a dechirper cavity then compensates the energy chirp by accelerating low-energy electrons while decelerating high-energy electrons, resulting in a narrowband energy-stabilized beam (**d**).

of a seeded free-electron laser from initially large energy spread laser-plasma electron beams[27], although scalability of the concept beyond a proof-of-principle experiment remains unclear.

Other techniques are based on passive structures to remove the energy chirp of a decompressed beam. In these dechirpers, the interaction of the electron bunch with a corrugated pipe[28], a dielectric structure[29] or a plasma[30–33] drives an electric field that effectively removes the correlated energy spread. However, because a passive dechirper is driven by the electron bunch itself, any small variation in bunch length, charge or current profile also affects the dechirping result, and as they also do not correct the beam energy jitter, stability concerns remain.

More recently, it has been proposed to add an accelerating, that is, active, structure after decompression[34–37] to greatly reduce both the correlated energy spread and energy jitter[38,39], thereby addressing short-comings of previous concepts and providing laser-plasma-generated electron beams of unprecedented quality and reproducibility.

In the following, we experimentally demonstrate, for the first time to our knowledge, active energy compression of a laser-plasma-generated electron beam. We improve the beam spectral properties by more than an order of magnitude and demonstrate performance previously only associated with modern RF accelerators.

Our energy compression scheme is illustrated in Fig. 1. A laser-plasma accelerator provides several-micrometres long, kiloampere peak current electron beams (Fig. 1a) of several-percent energy spread and energy jitter (Methods).

The laser-plasma accelerator is followed by a magnetic chicane. Here the first dipole introduces an energy-dependent deflection angle. The electron trajectories are then parallelized by a second dipole of inverse field. A third and fourth dipole close the symmetry and bring the beam back on the design axis. The chicane thereby introduces energy-dependent path-length differences that effectively stretch the bunch longitudinally (Fig. 1b) and induces an energy–position correlation (energy chirp).

After the chicane, the beam goes through an accelerating RF cavity, in which the positive gradient of the accelerating field cancels out the previously induced energy chirp (Fig. 1c). Through this mechanism, the set-up also removes the energy jitter: the electron spectrum is compressed to the electron energy that overlaps the zero crossing of the RF field (Fig. 1d).

Ideally, the beam energy spread is reduced proportionally to the bunch stretching, which can be more than two orders of magnitude. In practice, however, both the energy chirp introduced by the chicane and the sinusoidal RF field have small but non-negligible nonlinear terms that limit the energy compression. Yet, even including those nonlinear contributions, an energy spread reduction by more than an order of magnitude is readily possible (Methods). The scheme is thus ideally suited for short, high-current electron beams, as provided by a plasma accelerator, and applications that require only moderate peak current.

We have demonstrated this concept experimentally at the LUX laser-plasma accelerator. The drive laser provides 2.2-J, 35-fs (full width at half maximum (FWHM)) pulses on target at 1-Hz repetition rate. Through the interaction with a 5-mm-long plasma source, the set-up provides electron beams with an energy of 257 megaelectronvolts (MeV) at 41 pC (13 pC rms) of charge and a typical energy spread of 1.8% and energy jitter of 3.5%. From simulations, we estimate an initial bunch length of about 2 μm (rms) which corresponds to a peak current of 2.5 kA (Methods).

After the target, electrons are transported to the magnetic chicane, characterized by the chicane strength parameter $R_{56} = 100$ mm, which stretches 1% energy spread beams by a factor of about 1,000 to 1 mm length and induces an energy chirp of 1.0% per mm. The dechirper cavity (Methods) is a 5-m-long RF structure operated at 10-cm wavelength (S-band) and can change the beam energy by about 50 MeV.

After dechirping, electron beams are sent into a spectrometer and dispersed by a dipole magnet onto a scintillating screen to record the energy spectra with a resolution of order 0.07% (Methods). Not shown in Fig. 1, we have implemented several diagnostics throughout the set-up, including scintillating screens to measure the electron beam transverse profile and beam position monitors to non-invasively measure the transverse beam positions and charge (Methods).

First, we calculated the RF amplitude to remove the energy chirp, which is 45.4 MV (see Methods) for our chicane of $R_{56} = 100$ mm. We then scanned the RF phase to compress the electron beam energy (see Fig. 2).

As we scan the phase, the median electron beam energy follows the sinusoidal RF field (red dots).

At 0°, the bunch is centred at the zero crossing of the RF electric field (Fig. 2a). Electrons at the head of the bunch are decelerated, whereas electrons at the back of the bunch are accelerated, effectively reducing

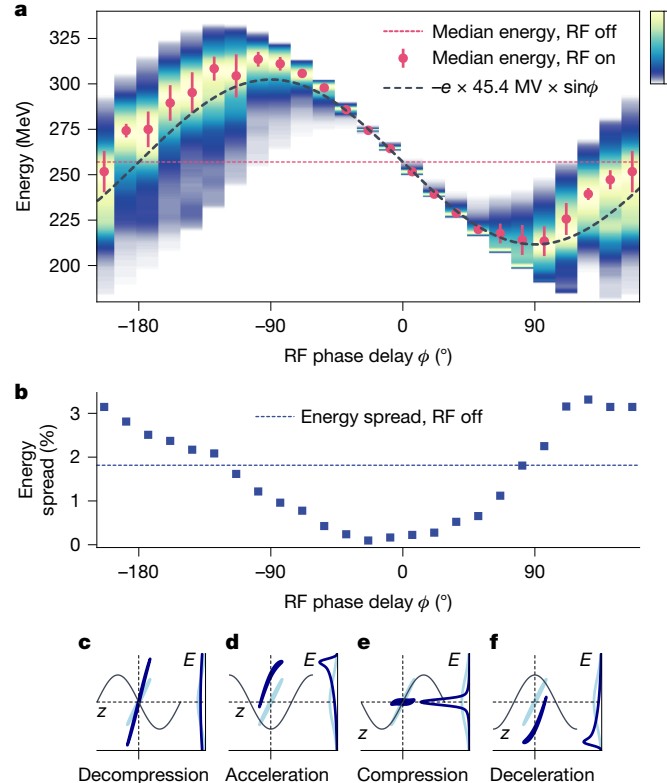

**Fig. 2 | Phase scan between laser-plasma electron beam and RF field. a**, Phase scan between electron bunch and RF field in steps of 15° (ref. 40). For better visibility, the spectral density is normalized for each step, averaging over 50 shots each. The shifted electron energy (red dots) follows the sinusoidal RF field (dashed black line). The energy jitter is denoted by red bars. **b**, At optimum compression, the energy spread is minimized. **c**–**f**, For illustration, we calculate the phase space of an initially chirped electron bunch (light blue) after interaction with the RF (dark blue) at distinct phases.

the chirp and, thus, energy spread. The opposite effect happens at a phase of ±180°, at which the slope of the RF is inverted: electrons at the head of the bunch are now accelerated, whereas electrons at the back of the bunch are decelerated, effectively increasing the chirp and broadening the spectrum. Around ±90°, the bunch is collectively decelerated and accelerated, respectively, which shifts the energy spectrum.

Notably, we find the smallest energy spread not at 0° but at a slightly shifted phase of −23.6° (Fig. 2b), which can be understood as follows. The second-order dispersion of the chicane adds a small curvature to the linear energy chirp. Therefore, we need to operate the RF slightly below 0°, at which the small curvature of the sinusoidal RF field just compensates the nonlinear chirp (Methods). Operating instead at 0°, the RF field is almost linear and cannot compensate the curvature of the chirp, resulting in a larger energy spread. In general, owing to these nonlinearities, there is a unique pair of amplitude and phase that can be calculated analytically (Methods) and results in the smallest possible energy spread.

At the optimum set point with minimum energy spread, we recorded about 1,000 shots with the RF turned off and on, shown in Fig. 3.

With the RF on (Fig. 3a), the energy jitter reduced by a factor of 72 from 3.5% to 0.048% and the energy spread reduced by a factor of 18 from 1.8% to 0.097%. Operating at a phase of −23.6°, the fully energy-compressed beam was shifted from a median energy of 257 MeV (RF off) to 275 MeV (RF on). The energy-compressed beams have a mean charge of 32 pC (12 pC rms). The peak spectral density reached as high as 70 pC per MeV. About 50% of all shots feature a sub-permille energy spread (Extended Data Fig. 1). Some shots were compressed to an energy spread as small as 0.068%, which is at the estimated resolution limit of our electron spectrometer. With compression, the charge provided inside a ±1% window of the median energy improved from 18.1% to 99.9%.

These results correspond to the best energy compression settings in the experiment, but we can further explore different capabilities of the set-up.

For example, stretching the electron bunch more lowers the energy chirp and thus reduces the required RF amplitude and thus power to operate the cavity. As the longer bunch then covers a larger phase of the RF, nonlinearities will reduce the energy compression performance. We tested this behaviour ($R_{56}$ = 170 mm, amplitude 28 MV) and could still reduce the energy jitter and spread to 0.09% and 0.13%, respectively, while consuming a factor of three less RF power.

Furthermore, we can vary the RF phase to fine-tune the target energy within a several-percent range without notable loss of compression performance.

In summary, our set-up provides electron beams with a performance in energy jitter and spread previously only obtained from modern RF accelerators, opening up widespread deployment of laser-plasma accelerator technology.

An important application for such an energy-compressed laser-plasma electron beam is an injector for a future synchrotron storage ring.

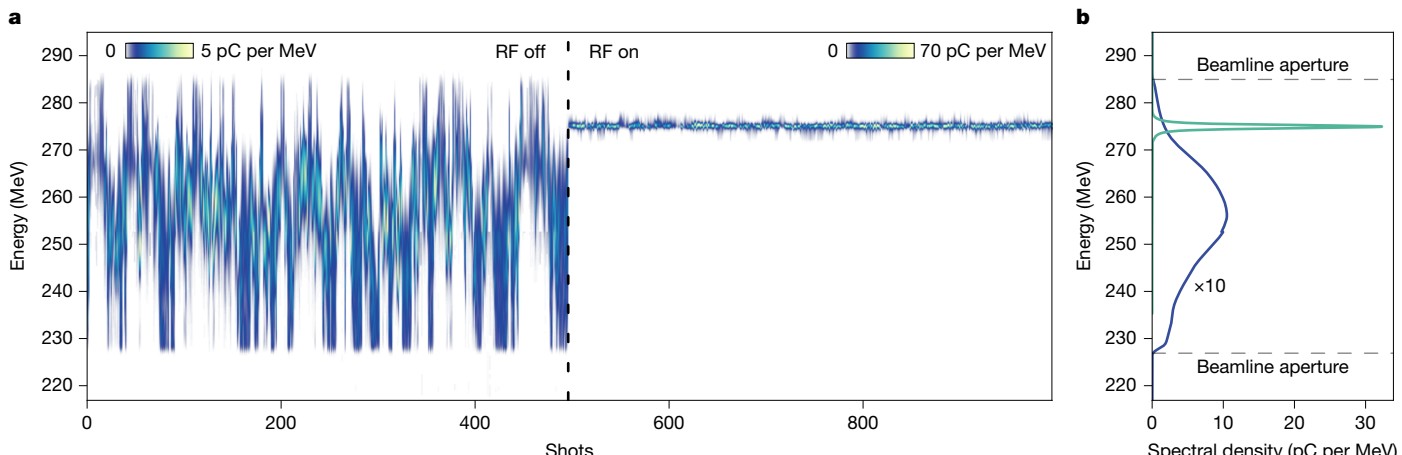

**Fig. 3 | Active energy compression. a**, Series of approximately 1,000 energy spectra on the electron spectrometer with RF off and on[40]. **b**, Average spectral density before (blue) and after (green) energy compression. The uncompressed spectrum is scaled by a factor of 10 for better visibility. The aperture of the beam pipe in the chicane defines a transmission window ranging from 227 to 285 MeV.

This application takes full advantage of the picosecond-level, several-ampere electron bunches of permille energy spread and jitter that our set-up already delivers today. Direct storage ring injection, as recently proposed[20], typically requires GeV-level electron beams. Recent work has already demonstrated laser-plasma accelerators delivering up to 10 GeV beam energy[5–9] using advanced laser guiding schemes to extend the interaction length of the drive laser and plasma. Furthermore, using X-band RF technology, the energy compression set-up could scale to higher beam energies without a marked increase in footprint[20,39]. With further development of high-efficiency, high-average-power laser drivers, a plasma-based injector could become a compact and energy-efficient alternative to RF technology[20].

Other applications requiring higher beam currents could use the stronger dechirping gradients of X-band RF technology[39] or plasmas[38] to improve the beam spectral properties while maintaining peak current.

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

## Methods

### Laser-plasma accelerator

The LUX laser-plasma accelerator has been previously described[22,41]. It is driven by the ANGUS Ti:sapphire laser providing 2.2-J (0.5% rms) pulse energy on target at 35-fs FWHM pulse length and a spot size of 22 µm FWHM diameter at 1-Hz repetition rate. Operational stability benefits greatly from the recently added new seed laser[42] and four InnoLas SpitLight 7000 pump lasers driving the last amplifier.

The laser is focused into a 5-mm-long, 500-µm-diameter channel continuously filled with hydrogen gas at 14.4 mbar. Electron beams are created at the beginning of the channel, at which an extra inlet, operated at 18.6 mbar and doped with 4.3% argon gas, creates a plasma density peak. Electron beams of roughly 41-pC (13 pC rms) charge are created at the transition region between the doped and undoped gas using the downramp-assisted localized ionization injection scheme[22,43] and then accelerated to about 257 MeV. Leaving the plasma, the electron beams feature a beam divergence of 1.1 mrad (0.9 mrad) and a pointing jitter of 0.6 mrad rms (0.4 mrad rms) in the horizontal (vertical) plane.

The set-up of drive laser, plasma accelerator and subsequent beamline is integrated into a control system to monitor the performance and record a complete set of beam properties and machine parameters for each individual event. Using Bayesian optimization[41,44], the plasma accelerator is routinely tuned to reproducible working points.

### Energy compression beamline

After the target, electron beams are captured by a pair of electromagnet quadrupoles and transported downstream. LUX was originally conceived to study undulator radiation. The energy compression experiment has been built as an extension to this beamline, which is bypassed for the measurements presented here.

For the energy compression experiment, electron beams are sent through a symmetric, four-dipole (212 mT, 9° deflection angle) C-shape chicane to stretch the bunch. For simplicity, the chicane was designed with a fixed stretching factor, $R_{56} = 100$ mm. The aperture of the beam pipe (vacuum chamber) in the dispersive section of the chicane limits the transmitted energy to approximately ±10% of the reference energy.

The chicane is followed by a 5-m-long S-band resonator cavity, operated at 2.998 GHz (10-cm wavelength). The cavity is powered by a pulse-forming network modulator and a klystron with a phase stability of 2° (peak to peak). The klystron provides up to 20 MW peak power pulses that are compressed to 80 MW (roughly a factor of 4) using a SLED cavity (0.4-µs pulse length). The experiment was designed with the RF equipment already available at our facility, which includes the RF cavity, klystron and modulator. Different RF technologies, for example, X-band, would support much higher beam energies while still keeping a compact footprint.

To synchronize the laser-plasma accelerator to the RF cavity, we use a joint 2.998-GHz master oscillator. The seed laser is stabilized in repetition rate and locked to the master oscillator. The klystron is also synchronized to the master oscillator. The phase between the RF cavity and laser-plasma accelerator is stabilized to 0.2° using a feedback loop.

The beamline features nine beam profile monitors and seven non-invasive cavity-type beam position monitors that also provide charge information and steering magnets to correct the beam position.

The electron spectrometer consists of a 0.9-T and 0.4-m-long dipole magnet surrounding a vacuum chamber. The dispersed electron beam leaves through one side of the triangular-shaped vacuum chamber wall, which is milled to a thickness of only 1 mm. The beam then passes through a scintillating screen mounted outside the chamber, at which it is recorded with a CCD camera. The spectrometer resolution at 300 MeV is about 0.07%, limited by the granularity of the scintillator, the pixel size of the CCD and the small amount of electron beam scattering as it passes through the spectrometer chamber wall.

### Statistical metrics for electron energy spectra

The electron spectra obtained from the laser-plasma accelerator can substantially deviate from a Gaussian distribution and can feature side-wings, which are typical for beam-loaded plasma acceleration[22]. As a result, the arithmetic mean and standard deviation are not reliable indicators of the centre and width of the energy distribution.

Instead, we use the median of the energy spectrum to provide a more meaningful measure for the beam central energy. We define the energy jitter as the median absolute deviation of all beam energies.

For the width of the spectrum, or energy spread, we use the relative median absolute deviation of the measured energy spectrum. To represent the typical energy spread of an ensemble of shots, we calculate the median of all energy spreads.

In case of a Gaussian distribution, the median absolute deviation is approximately 0.68 times the standard deviation.

### Simulations

Precise modelling of the LUX laser-plasma accelerator was developed in previous works[22,41,44] using the particle-in-cell code FBPIC[45] and Bayesian optimization. The simulations reproduce the experimental beam properties obtained in the optimal beam-loading regime over a wide range of parameters, which allows us to estimate certain beam parameters that are not directly measurable, such as the bunch length.

Beam tracking simulations through the energy compression beamline were performed with OCELOT[46], which uses up to second-order transfer maps to simulate beamline elements. The beams (modelled as sets of macroparticles) obtained from the FBPIC simulations are transported to OCELOT to complete start-to-end simulations of the whole set-up.

### Energy compression: analytical description

In the following, we analytically describe the dynamics of an electron beam within an energy compressor using longitudinal phase-space coordinates. Let $\delta = (E - E_{ref})/E_{ref}$ be the relative energy deviation and $\zeta = z - z_{ref}$ be the longitudinal position of the beam's electrons relative to a reference particle of energy $E_{ref}$ and position $z_{ref}$.

The chicane introduces an energy-dependent shift in the positions of the beam electrons. To the second order in $\delta_0$, this shift is given by

$$\zeta = \zeta_0 + R_{56}\delta_0 + T_{566}\delta_0^2, \tag{1}$$

in which $\zeta_0$ and $\delta_0$ are the initial coordinates of the electrons and $R_{56}$ and $T_{566}$ are the first-order and second-order longitudinal dispersion coefficients of the chicane, respectively. Through the chicane, to the first order in $\delta_0$, a beam with initial length $\sigma_{\zeta_0}$, initial spread $\sigma_{\delta_0}$ and no energy–position correlation is stretched to a length $\sigma_\zeta = R_{56}\sigma_{\delta_0}$. The uncorrelated energy spread becomes $\sigma_\delta^u = \sigma_{\zeta_0}/R_{56}$.

The RF cavity then applies a sinusoidal voltage to the electrons, given by $V_{RF}\sin(k_{RF}\zeta + \phi_0)$, in which $k_{RF} = 2\pi\nu_{RF}/c$ is the RF wavenumber, $\nu_{RF}$ is the frequency and $\phi_0$ is the offset of the RF signal relative to the position of the reference particle. The resulting energy deviation is

$$\delta = \delta_0 - \frac{eV_{RF}}{E_{ref}}\sin(k_{RF}\zeta + \phi_0), \tag{2}$$

which describes the final energy–position correlation (chirp) imparted to the beam by the energy compressor. Substituting equation (2) into equation (1), the conditions for dechirping that minimize the energy spread can be obtained by setting the first and second derivatives of $\delta(\zeta)$ to zero at $\zeta = 0$. The set of phase and RF amplitude yielding minimum energy spread is then

$$\tan(\phi_0) = \frac{2T_{566}}{k_{RF}R_{56}^2}, \tag{3}$$

$$eV_{RF} = \frac{E_{ref}}{k_{RF}R_{56}} \sqrt{1 + \left(\frac{2T_{566}}{k_{RF}R_{56}^2}\right)^2}. \tag{4}$$

In a dipole chicane, $T_{566} \simeq -(3/2)R_{56}$, which simplifies $\tan(\phi_0) \simeq -3/(k_{RF}R_{56})$. For $R_{56} = 100$ mm, $\nu_{RF} = 3$ GHz and $E_{ref} = 257$ MeV, this gives $\phi_0 = -25.5°$ and $V_{RF} = 45.2$ MV.

Despite fulfilling the dechirping conditions of equations (3) and (4), the electron beam will still retain a finite correlated energy spread owing to nonlinearities in equation (2) away from zero. For an initially Gaussian-distributed beam centred on $E_{ref}$, the resulting correlated energy spread is estimated to leading order as

$$\sigma_\delta^c \simeq 0.645(k_{RF}R_{56})^2\sigma_{\delta_0}^3. \tag{5}$$

The final energy spread is given by the quadratic sum of the correlated and uncorrelated components.

Applying equation (5) with the experimental parameters and assuming an initial energy spread of 3.0%, the final correlated energy spread after dechirping is estimated to be 0.07%, which is already at the resolution limit of the spectrometer. Given the short, micron-scale length of the laser-plasma electron beams, the uncorrelated energy spread can reach levels near 0.001%, which could be exploited by new applications[47].

## Energy compression: transverse beam properties and charge performance

The beam optic increases the estimated 5-μm beam size at the plasma exit to about 0.5 mm at the RF cavity. The beam pointing (directional) jitter at the plasma source translates into a transverse position jitter at the RF cavity, which is on the same order as the transverse beam size and much smaller than the good-field region of the cavity field. The geometric aperture of the cavity is 22 mm (diameter). Therefore, electron beam pointing jitter has negligible impact on the energy compression performance.

At the exit of the plasma source, we estimate a normalized emittance of about 3 μm, corresponding to a geometric emittance of approximately 5 nm, consistent with measurements and the typical performance of the plasma source[22].

As the percent-level energy spread beam is leaving the target, its divergence causes a chromatic emittance degradation. This effect can be mitigated with a further small chromatic chicane[20,39].

For this proof-of-principle experiment, we have not installed such a chromatic chicane. Without it, the emittance grows to order 10 μm (20 nm) normalized (geometric) emittance after the energy compression set-up. The resulting beams would, for example, fit well within the 10-μm geometric acceptance of the DESY II booster synchrotron.

We measure a mean beam charge of 38 pC at the beginning of the energy compression set-up, 33 pC after the decompression chicane (89% transmission) and 32 pC after the RF cavity (96% transmission), resulting in an overall transmission of 85% through the set-up. We attribute the loss in charge mainly to the limited beam pipe aperture in the chicane section.

The charge stability of our laser-plasma source can be mainly attributed to the reproducibility in wavefront of our drive laser[22], but is not fundamentally limited.

## Data availability

The data that support the findings of this study are available from Zenodo[40] with the identifier https://doi.org/10.5281/zenodo.14762556.

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

**Acknowledgements** We especially thank R. Jonas for installing the waveguides, H. Ehrlichmann for suggesting S-band technology and L. Heddendorp for engineering support and coordination. We thank all workshops and technical groups at DESY, especially MIN, MEA, MVS, MSK, MPC, MKK, MCS, MPS, D3, MIL and MDI, for support. This work was supported by Helmholtz ARD, the Maxwell computational resources at DESY and received financing from Deutsche Forschungsgemeinschaft (DFG, German Research Foundation) – 491245950.

**Author contributions** A.M.d.lO., I.A., S.A.A., R.B., A.F.P. and M.Th. developed the concept for active energy compression of a laser-plasma accelerator. P.W. and A.R.M. conceived the experimental realization, with input from all co-authors and DESY technical groups. P.W., M.Tr., L.H. and S.A.A. designed the beamline. K.S. and P.A.W. developed safety concepts and provided project management support. P.W., M.Tr., L.H., S.J. and M.K. built the experiment, with help from DESY technical groups. T.E., T.H., G.P., M.S. and C.W. upgraded and operated the laser. P.W., M.Tr., L.H., S.J., M.K. and A.R.M. performed the experiment. P.W., L.H., A.M.d.lO., S.J. and M.K. analysed the data. W.P.L. and A.R.M. provided guidance to the project. P.W., M.Tr., L.H., A.M.d.lO., S.J., M.K., W.P.L. and A.R.M. wrote the manuscript.

**Funding** Open access funding provided by Deutsches Elektronen-Synchrotron (DESY).

**Additional information**

**Correspondence and requests for materials** should be addressed to P. Winkler or A. R. Maier.

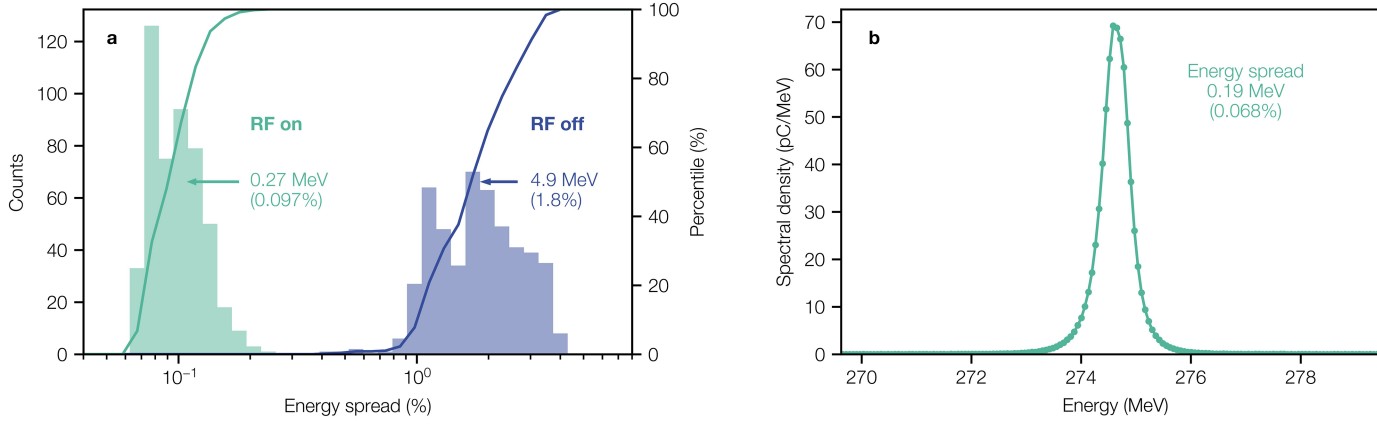

**Extended Data Fig. 1 | Improvement of energy spread performance. a**, Distribution of energy spreads. With the RF turned on, the energy spread improves from 1.8% (indicated in blue) to 0.097% (indicated in green). Arrows denote the 50th percentile for both cases. **b**, Individual energy-compressed shots reach an energy spread as low as 0.068%.