## [Peer Review file · Nature]

Active energy compression of a laser-plasma electron beam

Corresponding Author: Dr Paul Winkler

Version 0:

Reviewer comments:

Referee #1

(Remarks to the Author)

The manuscript titled 'Energy compression of a laser-plasma-accelerator' by P. Winkler et al. presents a solution to the persistent challenge to combine laser wakefield accelerators with classical accelerator beamline components. Central to any solution is the management of the comparably large energy spread of plasma accelerators that has to be effectively mitigated in order to avoid issues like chromatic emittance growth or straight out energy acceptance of machines like synchrotron light sources, for which the DESY team is planning to eventually replace the classical injector beam line. Apparently, reaching this goal would have enormous impact in the applied accelerator and light source community. In a proof-of-concept experiment the team employs classical phase space manipulation elements like a magnetic chicane for stretching (de-compression) of the inherently short electron pulse followed by an rf acceleration cavity that reduces the energy spread of the now long pulse and thus translates a short and broad band bunch into a long and narrow energy spread one. It is interesting to see that at this point also higher order components become visible and can be compensated for. It should also be noted that already the quality of the original electron bunch is of leading quality guaranteed by machine learning based optimization routines. Results presented are absolutely convincing and confirm the expected potential of the method, at least at the electron energy of around 250 MeV that was used and I would like to congratulate the authors to this achievement.

In general, the manuscript is very well written, in particular in addressing the broad readership of nature as I would have expected from this group. It focuses on the key story line and all details of interest for the specialists are adequately provided in the methods section.

Now the difficult question remains if these undoubtedly excellent results warrant publication in nature. One could argue that the techniques applied are kind of text-book accelerator physics techniques and have in part been applied in different context at advanced accelerators (as the authors correctly cite) already. One example is the (identical) decompression applied to the electron bunch to improve slice properties for the driving of FELs.

Another (not cited) is that a related concept (though different in its realization as beams had been non-relativistic and thus time-of-flight stretching happened naturally) had been applied to recompress and / or energy-compress laser accelerated proton beams [Scientific Reports 5, 12459 (2015)] also with the long-term prospect to perform injection into the heavy ion synchrotron SIS at GSI. Similar concepts based on plasma based techniques instead of rf ones are discussed by the authors, but one can argue that these typically used conventional injectors. One point I would have to make is that though the proof-of-concept is well realized, injection into a modern synchrotron would typically require energies of many GeV

and the paper so far lacks an at least brief discussion on how this demand would affect the solution (i.e. is the then required rf cavity still size and cost effective).

In total, as this is sort of a machine physics type of demonstration experiment at a rather low energy and as it lacks a truly innovative concept, I am inclined to recommend publication in a more specialized high impact journal.

In any case I have a few suggestions the authors might want to consider:

- 1) I would recommend to include the reference to the proton experiment I mentioned.
- 2) Figure 2 nicely presents the effects of scanning the phase of the rf over the full cycle. This looks nice and is also convincing, but the scales used reduce the visibility of the effects of interest, namely the energy spread in the case of best performance. It might also be of interest to show how the actual distribution looks like if not only representing spectrometer resolution. Maybe an additional panel or a zoom into the optimal region could help.
- 3) In line 317 (methods) the authors mention scattering in the vacuum chamber of the spectrometer. It is not clear to me where the beam is scattering from.

Referee #2

(Remarks to the Author)

Review of Nature manuscript 2024-11-24561

This paper presents a scheme for stabilizing the median energy and significantly narrowing the energy spread of laser wakefield accelerated electrons. The method uses a magnetic chicane to chirp the electron bunch followed by an RF cavity for dechirping.

The paper is well written, with the background physics and supporting citations clearly laid out. The results are impressive and are potentially a milestone in making laser plasma accelerator beams of sufficiently high quality for applications such as free electron lasers.

There are two areas that need some discussion. One is beam pointing stability and divergence--- properties of high importance when considering laser wakefield accelerated beams for applications. What is the effect of this scheme on those parameters? The other is a discussion of how this scheme might scale to laser wakefield accelerated electron beams of several GeV energy (an order of magnitude higher than here), which are now being generated more easily and frequently by groups worldwide— such energies are of special interest for FEL injectors, for example. Can the chicane/dechirper method be applied directly to multi-GeV bunches, and if so, what kinds of equipment and parameters are needed? Or is the 0.275 GeV beam of this paper a candidate for injection into a laser-driven acceleration stage?

Referee #3

(Remarks to the Author)

This paper describes an experimental demonstration of an RF-based energy compression scheme for laser wakefield accelerated (LWFA) electron beams. The scheme is based on the combination of a dispersive magnetic chicane, to impart longitudinal position-energy correlations, and then a RF-cavity to provide the energy compression. The work follows on from numerical work by Antipov et al. and Ferran Pousa et al. (both referenced in the manuscript) and opens the way for the use of LWFA as an injector for electron synchrotrons.

The work is highly significant and novel and is sure to attract a high level of interest from the accelerator community and beyond. Plasma acceleration has long promised to be an alternative to RF- based technology but has yet to become part of the mainstream. This experiment shows that LWFA can achieve performance levels of interest to large accelerator facilities. There are still some further improvements required to achieve the full specifications of a synchrotron injector, (e.g. energy level, average beam current and reliability) but this experimental demonstration overcomes arguably the most important scientific challenge and so should greatly increase the confidence in the future of plasma acceleration.

The quality of the methods, the data and the overall presentation is excellent. The treatment of statistics and uncertainty is appropriate, and I have no doubts over the results. The conclusions appear valid and robust, and, to my knowledge, the relevant previous works have been cited and discussed.

I have the follow comments that I would like the authors to address:

- 1) In the introduction (line 34) the authors use the term Laser Plasma Acceleration (LPA) to refer specifically to laser-drive

plasma wakefield acceleration – commonly referred to as LWFA. As there are many different types of laser plasma acceleration (not just LWFA) I suggest this is changed to the more specific LWFA term. LPA is also used in the abstract, but this could be seen as a higher-level conceptual description and so could be left as is.

2) On line 59, the authors refer to ‘spurious chromatic effects’, whereas I believe they mean ‘adverse chromatic effects’ or something equivalent.

3) Figure 3 demonstrates the remarkable experimental performance in narrowing the beam spectrum and stabilizing the mean energy. I would also like there to be a discussion of the charge performance of the system, and the shot-to-shot charge variation of the final beam. There is a comment about 99.9% of the charge in the final beam is within a 2% window, but it is not clear if any charge is lost during the beam transport. A plot showing the total charge vs shot number would be beneficial in further understanding the results.

4) Similarly, there is no comment on the transverse beam quality. Does this match expectations (e.g. from references 36,37)? As emittance and charge are key requirements for an injector, I would expect these quantities to be reported and discussed.

Reply to Reviewer Comments

Active energy compression of a laser-plasma electron beam

P. Winkler et al.

February 4th, 2025

We thank all the reviewers for their valuable feedback. In the following, we provide a detailed answer to their comments.

Reviewer #1

Reviewer comment:

The manuscript titled 'Energy compression of a laser-plasma-accelerator' by P. Winkler et al. presents a solution to the persistent challenge to combine laser wakefield accelerators with classical accelerator beamline components. Central to any solution is the management of the comparably large energy spread of plasma accelerators that has to be effectively mitigated in order to avoid issues like chromatic emittance growth or straight out energy acceptance of machines like synchrotron light sources, for which the DESY team is planning to eventually replace the classical injector beam line. Apparently, reaching this goal would have enormous impact in the applied accelerator and light source community. In a proof-of-concept experiment the team employs classical phase space manipulation elements like a magnetic chicane for stretching (de-compression) of the inherently short electron pulse followed by an rf acceleration cavity that reduces the energy spread of the now long pulse and thus translates a short and broad band bunch into a long and narrow energy spread one. It is interesting to see that at this point also higher order components become visible and can be compensated for. It should also be noted that already the quality of the original electron bunch is of leading quality guaranteed by machine learning based optimization routines. Results presented are absolutely convincing and confirm the expected potential of the method, at least at the electron energy of around 250 MeV that was used and I would like to congratulate the authors to this achievement.

In general, the manuscript is very well written, in particular in addressing the broad readership of nature as I would have expected from this group. It focuses on the key story line and all details of interest for the specialists are adequately provided in the methods section.

Now the difficult question remains if these undoubtedly excellent results warrant publication in nature. One could argue that the techniques applied are kind of textbook accelerator physics techniques and have in part been applied in different context at advanced accelerators (as the authors correctly cite) already. One example is the (identical) decompression applied to the electron bunch to improve slice properties for the driving of FELs. Another (not cited) is that a related concept (though different in its realization as beams had been non-relativistic and thus time-of-flight stretching

happened naturally) had been applied to recompress and / or energy-compress laser accelerated proton beams [Scientific Reports 5, 12459 (2015)] also with the long-term prospect to perform injection into the heavy ion synchrotron SIS at GSI. Similar concepts based on plasma based techniques instead of rf ones are discussed by the authors, but one can argue that these typically used conventional injectors. One point I would have to make is that though the proof-of-concept is well realized, injection into a modern synchrotron would typically require energies of many GeV and the paper so far lacks an at least brief discussion on how this demand would affect the solution (i.e. is the then required rf cavity still size and cost effective).

In total, as this is sort of a machine physics type of demonstration experiment at a rather low energy and as it lacks a truly innovative concept, I am inclined to recommend publication in a more specialized high impact journal.

We would like to thank the reviewer for this overall positive assessment of the quality of our work. We have carefully revised our manuscript based on the reviewer's comments.

Following the reviewer's suggestion, we have expanded the discussion section to provide more information on the scalability of the concept towards GeV-level beam energies. The relevant paragraph now reads:

“A prime application for such an energy compressed laser-plasma electron beam is an injector for a future synchrotron storage ring. This application takes full advantage of the picosecond-level, few-ampere electron bunches of permille energy spread and jitter, that our setup delivers already today. Direct storage ring injection, as recently proposed²⁰, typically requires GeV-level electron beams. Recent work has already demonstrated laser-plasma accelerators delivering up to 10 GeV beam energy^{5,6,7,8,9} using advanced laser guiding schemes to extend the interaction length of drive laser and plasma. Furthermore, using X-band RF technology, the energy compression setup could scale to higher beam energies without significant increase in footprint^{20, 39}. With further development of high-efficiency, high average power laser drivers, a plasma-based injector could become a compact and energy-efficient alternative to RF technology²⁰.”

The two main ingredients required to scale our concept to the GeV-level beam energies required for direct synchrotron injection are:

First, increasing the interaction length of laser and plasma by means of active drive laser pulse guiding. The community has demonstrated rapid progress in this area, with

the most recent results, Ref. 9, reaching close to 10 GeV electron beam energies. While our proof-of-principle setup did not include active laser guiding and used drive pulse of comparably low energy, we are confident that the required GeV-level electron beams are, in principle, available.

Second, using active cavities of higher gradients to keep a compact footprint. Our proof-of-concept experiment was built with hardware that was readily available to us, and as a result we used S-band RF technology. We have, however, presented a more compact design based on X-band RF technology, compare Ref. 39, and also presented the concept of a plasma-based de-chirper cavity that could be built even more compact, compare Ref. 38.

The design for a 6 GeV laser-plasma-based injector for the future PETRA IV synchrotron is described in Ref. 20. The concept is based on, first, hydrodynamic, optically-field-ionized (HOFI) plasma channels to guide the laser pulse to deliver 6 GeV laser-plasma electron beams and, second, a 5-m-long active X-radio-frequency cavity to de-chirp the electron beam within a short distance.

In conclusion, we think that our concept scales well to GeV-level electron beam energies. For better readability, we expand the discussion section as presented above, and cite Refs. 20 and 39 for further information.

In the following, we address the other specific suggestions by the reviewer.

Reviewer comment:

In any case I have a few suggestions the authors might want to consider:

1) I would recommend to include the reference to the proton experiment I mentioned.

We have added the reference as requested.

Reviewer comment:

2) Figure 2 nicely presents the effects of scanning the phase of the rf over the full cycle. This looks nice and is also convincing, but the scales used reduce the visibility of the effects of interest, namely the energy spread in the case of best performance. It might also be of interest to show how the actual distribution looks like if not only

representing spectrometer resolution. Maybe an additional panel or a zoom into the optimal region could help.

We would like to thank the reviewer for this comment. We have very carefully considered the suggestion by the reviewer to expand Figure 2. However, in the end, we have come to the conclusion that we would prefer to not change the composition of the figure.

Following the conceptual illustration of the energy compression setup in Figure 1, the intended purpose of Figure 2 is to introduce – with measurements – the basic principles of the energy compression concept, while a more detailed view is then provided in Figure 3.

For that reason, we intentionally chose a scale for Figure 2 that emphasizes the main mechanism of compressing the energy spectrum as we change the phase. We are concerned that adding another panel would distract the reader from that main story-line. Yet, more detailed information on the energy spectrum in case of best performance is provided in Figure 3, and we provide a single-shot example of best performance in Extended Data Figure 1.

To further address the reviewer's comment, we have made our data publicly available via zenodo, compare Ref. 40, including that of Figure 2. We hope this will enable the interested reader to study the data in more detail.

Reviewer comment:

3) In line 317 (methods) the authors mention scattering in the vacuum chamber of the spectrometer. It is not clear to me where the beam is scattering from.

The scintillating screen to record the dispersed electron beam is mounted on-air, onto the spectrometer vacuum chamber. As the electron beams passes the vacuum chamber wall a small amount of scattering happens, which very slightly increases the beam size and thus limits to spectrometer resolution. We have extended the Methods section accordingly:

“The electron spectrometer consists of a 0.9 T and 0.4 m-long dipole magnet surrounding a vacuum chamber. The dispersed electron beam leaves through on side of the triangular-shaped vacuum chamber wall, which is milled to a thickness of only 1 mm. The beam then passes through a scintillating screen mounted outside to the chamber, where it is recorded with a CCD camera. The spectrometer resolution at 300 MeV is about 0.07 %, limited by the granularity of the scintillator, the pixel size of

the CCD, and the small amount of electron beam scattering as it passes through the spectrometer chamber wall.”

Once again, we would like to thank the reviewer for the valuable feedback.

Reviewer #2

Reviewer comment:

This paper presents a scheme for stabilizing the median energy and significantly narrowing the energy spread of laser wakefield accelerated electrons. The method uses a magnetic chicane to chirp the electron bunch followed by an RF cavity for dechirping.

The paper is well written, with the background physics and supporting citations clearly laid out. The results are impressive and are potentially a milestone in making laser plasma accelerator beams of sufficiently high quality for applications such as free electron lasers.

We would like to thank the reviewer for this positive feedback on our manuscript.

Reviewer comment:

There are two areas that need some discussion. One is beam pointing stability and divergence--- properties of high importance when considering laser wakefield accelerated beams for applications. What is the effect of this scheme on those parameters?

We thank the reviewer for this feedback. Following the reviewer's comment, we have added information on beam divergence to the Methods section:

“Leaving the plasma, the electron beams feature a beam divergence of 1.1 mrad (0.9 mrad) and a pointing jitter of 0.6 mrad rms (0.4 mrad rms) in the horizontal (vertical) plane.”

The beam size at the exit of the plasma source is estimated to order 5 μm . Our beam optic is set up such that it increases the beam size at the RF cavity to order of 0.5 mm. Furthermore, the pointing (directional) jitter at the plasma source translates into a transverse position jitter at the RF cavity, which is, however, on the same order than the transverse beam size and significantly smaller than the good-field region of the cavity field. Note, that the geometric aperture of the cavity is 22 mm (diameter). Thereby, the pointing jitter of the electron beam has negligible impact on the energy compression performance.

In principle, the energy compressor itself preserves the transverse beam quality. We do, however, expect some emittance growth due to chromatic effects during beam capture right after the plasma target. These effects on the emittance can be mitigated using an additional chromatic chicane as described in Ref. 39.

We have expanded the methods section with a brief discussion of these aspects:

“The beam optic increases the estimated 5 μm beam size at the plasma exit to order 0.5 mm at the RF cavity. The beam pointing (directional) jitter at the plasma source translates into a transverse position jitter at the RF cavity, which is on the same order than the transverse beam size and significantly smaller than the good-field region of the cavity field. The geometric aperture of the cavity is 22 mm (diameter). Therefore, electron beam pointing jitter has negligible impact on the energy compression performance.”

Note that this comment by reviewer #2 is similar to a comment by reviewer #3 addressing the effects on transverse beam quality. We provide more details and a discussion on beam emittance in our reply to reviewer #3.

Reviewer comment:

The other is a discussion of how this scheme might scale to laser wakefield accelerated electron beams of several GeV energy (an order of magnitude higher than here), which are now being generated more easily and frequently by groups worldwide— such energies are of special interest for FEL injectors, for example. Can the chicane/dechirper method be applied directly to multi-GeV bunches, and if so, what kinds of equipment and parameters are needed?

We would like to thank the reviewer for this feedback.

This comment is very similar to a comment raised by reviewer #1. We have expanded the discussion section to address the scaling of the energy compression setup.

For convenience, we repeat below some of the arguments addressing reviewer #1.

The two main ingredients required to scale our concept to the GeV-level beam energies required for direct synchrotron injection are:

First, increasing the interaction length of laser and plasma by means of active drive laser pulse guiding. The community has demonstrated rapid progress in this area, with

the most recent results, Ref. 9, reaching close to 10 GeV electron beam energies.

While our proof-of-principle setup did not include active laser guiding and used drive pulse of comparably low energy, we are confident that the required GeV-level electron beams are, in principle, available.

Second, using active cavities of higher gradients to keep a compact footprint. Our proof-of-concept experiment was built with hardware that was readily available to us, and as results we used S-band RF technology. We have, however, presented a more compact design, using X-band RF technology, compare Ref. 39, and also presented the concept a plasma-based de-chirper cavity that could be built even more compact, compare Ref. 38.

The design for a 6 GeV laser-plasma-based injector for the future PETRA IV synchrotron is described in Ref. 20. The concept is based on, first, hydrodynamic, optically-field-ionized (HOFI) plasma channels to guide the laser pulse to deliver 6 GeV laser-plasma electron beams and, second, a 5-m-long active X-radio-frequency cavity to de-chirp the electron beam within a short distance.

The revised discussion section now reads:

“A prime application for such an energy compressed laser-plasma electron beam is an injector for a future synchrotron storage ring. This application takes full advantage of the picosecond-level, few-ampere electron bunches of permille energy spread and jitter, that our setup delivers already today. Direct storage ring injection, as recently proposed²⁰, typically requires GeV-level electron beams. Recent work has already demonstrated laser-plasma accelerators delivering up to 10 GeV beam energy^{5,6,7,8,9} using advanced laser guiding schemes to extend the interaction length of drive laser and plasma. Furthermore, using X-band RF technology, the energy compression setup could scale to higher beam energies without significant increase in footprint^{20, 39}. With further development of high-efficiency, high average power laser drivers, a plasma-based injector could become a compact and energy-efficient alternative to RF technology²⁰.”

To achieve energy spread and jitter suitable for direct synchrotron injection, the setup requires laser-plasma electron beams of percent-level energy spread and jitter as an input. We have demonstrated this performance in our own experiment and see no fundamental reason, why laser-plasma accelerators at higher beam energies should not perform comparably.

In conclusion, we think that our concept scales well to GeV-level electron beam energies.

Reviewer comment:

Or is the 0.275 GeV beam of this paper a candidate for injection into a laser-driven acceleration stage?

This is a very interesting comment raised by reviewer #2. In Ref. 39, we describe a plasma based de-chirper cavity that compresses the energy spectrum, while still keeping a comparably short electron bunch length. With such short bunches, injection into a sub-sequent plasma stage to further boost the electron beam energy should, in principle, be possible. Aspects like beam loading and energy spread conservation in such a plasma booster stage would need to be carefully studied. This is, however, beyond the scope of this current manuscript.

Once again, we would like to thank the reviewer for the valuable feedback.

Reviewer #3

Reviewer comment:

This paper describes an experimental demonstration of an RF-based energy compression scheme for laser wakefield accelerated (LWFA) electron beams. The scheme is based on the combination of a dispersive magnetic chicane, to impart longitudinal position-energy correlations, and then a RF-cavity to provide the energy compression. The work follows on from numerical work by Antipov et al. and Ferran Pousa et al. (both referenced in the manuscript) and opens the way for the use of LWFA as an injector for electron synchrotrons.

The work is highly significant and novel and is sure to attract a high level of interest from the accelerator community and beyond. Plasma acceleration has long promised to be an alternative to RF-based technology but has yet to become part of the mainstream. This experiment shows that LWFA can achieve performance levels of interest to large accelerator facilities. There are still some further improvements required to achieve the full specifications of a synchrotron injector, (e.g. energy level, average beam current and reliability) but this experimental demonstration overcomes arguably the most important scientific challenge and so should greatly increase the confidence in the future of plasma acceleration.

The quality of the methods, the data and the overall presentation is excellent. The treatment of statistics and uncertainty is appropriate, and I have no doubts over the results. The conclusions appear valid and robust, and, to my knowledge, the relevant previous works have been cited and discussed.

We would like to thank the reviewer for this positive feedback on our manuscript.

Reviewer comment:

I have the follow comments that I would like the authors to address:

1) In the introduction (line 34) the authors use the term Laser Plasma Acceleration (LPA) to refer specifically to laser-drive plasma wakefield acceleration – commonly referred to as LWFA. As there are many different types of laser plasma acceleration (not just LWFA) I suggest this is changed to the more specific LWFA term. LPA is also

used in the abstract, but this could be seen as a higher-level conceptual description and so could be left as is.

We thank the reviewer for this feedback. As we use the abbreviation “LPA” only a few times throughout the whole manuscript, we decided to not use any abbreviation at all, neither LPA nor LWFA. Instead, for better readability, we have revised the manuscript and removed abbreviations where possible replacing them with more specific terms.

Reviewer comment:

2) On line 59, the authors refer to ‘spurious chromatic effects’, whereas I believe they mean ‘adverse chromatic effects’ or something equivalent.

We would like to thank the reviewer for this comment and have corrected the manuscript accordingly.

Reviewer comment:

3) Figure 3 demonstrates the remarkable experimental performance in narrowing the beam spectrum and stabilizing the mean energy. I would also like there to be a discussion of the charge performance of the system, and the shot-to-shot charge variation of the final beam.

There is a comment about 99.9% of the charge in the final beam is within a 2% window, but it is not clear if any charge is lost during the beam transport. A plot showing the total charge vs shot number would be beneficial in further understanding the results.

We would like to thank the reviewer for this important comment. We have added the mean charge and rms charge stability directly after the plasma target and after the energy compression setup to the manuscript (main text):

“Through the interaction with a 5 mm-long plasma source, the setup provides electron beams with an energy of 257 MeV at 41 pC (13 pC rms) of charge and ... “

“The energy-compressed beams have a mean charge of 32 pC (12 pC rms).”

LUX was originally conceived to study undulator radiation. The energy compression experiment has been built as an extension to this beamline, which is bypassed for the measurements presented here.

We measure a mean beam charge of 38 pC at the beginning of the energy compression setup, 33 pC after the decompression chicane (89% transmission), and 32 pC after the RF cavity (96% transmission), resulting in an overall transmission of 85% through the setup. We attribute the loss in charge mainly to the limited beam pipe aperture in the chicane section.

To further address the reviewer's comment, we make our data for Figures 2 and Figure 3 publicly available and have also included the charge on a shot-to-shot basis.

In the experiment, the charge stability of our laser-plasma source was of order 30% rms, which can be mainly attributed to the reproducibility in wavefront of our drive laser, compare Ref. 22. However, the charge stability it is not fundamentally limited, and significant improvements can be expected in the future.

In Ref. 20, we discuss the application of an energy-compressed laser-plasma-driven electron beam as a synchrotron injector. We show that using appropriate filling schemes, a 10% charge variation is compatible with top-up operation of a synchrotron storage ring. The main reason is, that a plasma-based injector would operate at slightly lower charge per bunch but at higher repetition rate compared to an RF-based injector. With a suitable filling scheme, the variations on bunch charge average out quickly.

All referees have stated that we separated very well information for the general reader (main text) from information relevant only for the specialist (methods). For the sake of readability, we therefore provide this additional information in a new methods section "Energy compression: transverse beam properties and charge performance".

Reviewer comment:

4) Similarly, there is no comment on the transverse beam quality. Does this match expectations (e.g. from references 38,39)? As emittance and charge are key requirements for an injector, I would expect these quantities to be reported and discussed..

We thank the reviewer for this important comment.

We estimate a normalized emittance of order 3 μm , corresponding to a geometric emittance of order 5 nm. These values are consistent with measurements and the typical performance of our plasma source, compare Ref. 22.

As the percent-level energy spread beam is leaving the target, its divergence causes a chromatic emittance degradation. To compensate this emittance growth, we have previously proposed to install an additional small chromatic chicane directly after the target, compare Ref. 39, and before the actual decompression chicane. This concept does not only mitigate chromatic emittance growth, but also mitigate coherent synchrotron radiation (CSR) effects in the decompression chicane.

For simplicity, we have not installed such a chicane in this proof-of-principle experiment. Without this chicane, the emittance grows to order $10 \mu\text{m}$ normalized emittance (20 nm geometric emittance) after the energy compression setup.

To put the numbers into context, these 20 nm geometric emittance beams would fit well within the $10 \mu\text{m}$ geometric acceptance of the DESY II booster ring, which accepts 400 MeV beam and boost them to 6 GeV for injection into the PETRA III storage ring light source.

We have extended the methods section “Energy compression: transverse beam properties and charge performance” to provide this additional information

Once again, we would like to thank the reviewer for the valuable feedback.